# Play to the Score: Stage-Guided Dynamic Multi-Sensory Fusion for Robotic Manipulation

**Ruoxuan Feng[1]**     **Di Hu[1,*]**     **Wenke Ma[2]**     **Xuelong Li[3]**

[1]Gaoling School of Artificial Intelligence, Renmin University of China
[2]Shenzhen Taobotics Co., Ltd.
[3]Institute of Artificial Intelligence (TeleAI), China Telecom
`{fengruoxuan, dihu}@ruc.edu.cn, wenke@taobotics.com, xuelong_li@ieee.org`

**Abstract:** Humans possess a remarkable talent for flexibly alternating to different senses when interacting with the environment. Picture a chef skillfully gauging the timing of ingredient additions and controlling the heat according to the colors, sounds, and aromas, seamlessly navigating through every stage of the complex cooking process. This ability is founded upon a thorough comprehension of task stages, as achieving the sub-goal within each stage can necessitate the utilization of different senses. In order to endow robots with similar ability, we incorporate the task stages divided by sub-goals into the imitation learning process to accordingly guide dynamic multi-sensory fusion. We propose MS-Bot, a stage-guided dynamic multi-sensory fusion method with coarse-to-fine stage understanding, which dynamically adjusts the priority of modalities based on the fine-grained state within the predicted current stage. We train a robot system equipped with visual, auditory, and tactile sensors to accomplish challenging robotic manipulation tasks: pouring and peg insertion with keyway. Experimental results indicate that our approach enables more effective and explainable dynamic fusion, aligning more closely with the human fusion process than existing methods. The code and demos can be found at https://gewu-lab.github.io/MS-Bot/.

**Keywords:** Multi-Sensory, Robotic Manipulation, Multi-Stage

## 1 Introduction

Humans are blessed with the ability to flexibly use various sensors to perceive and interact with the world. Over the years, endowing robots with this potent capability has remained a dream for humanity. Fortunately, with the advancement of computing devices and sensors [1, 2], building a multi-sensory robot system to assist humans has gradually become achievable. We have witnessed the emergence of some exciting multi-sensory robots, such as *Optimus* [3] and *Figure 01* [4].

One challenge in tasks that robots assist humans with is the complex object manipulation task, which requires achieving a series of sub-goals. These sub-goals naturally divide a complex task into multiple stages. Many efforts attempt to make models comprehend these task stages, whether through hierarchical learning [5, 6, 7] or employing LLMs [8, 9, 10]. However, when incorporated into multi-sensory robots, this challenge becomes more profound. We need to not only understand the stages themselves but also rethink multi-sensory fusion from the fresh perspective of task stages.

Completing sub-goals within a complex task may require different senses, with varying modality importance at each stage. Thus, stage transitions may entail changes in modality importance. To illustrate this, we build a multi-sensory robot system (Fig. 1, left). We then use imitation learning to train a MULSA [11] model with self-attention fusion to complete pouring task and record the testing confidence (Fig. 1, right). We observe correlations between the confidence and task stages after

---

* Corresponding author. "Play to the score" is slang for adjusting actions to fit current circumstances.

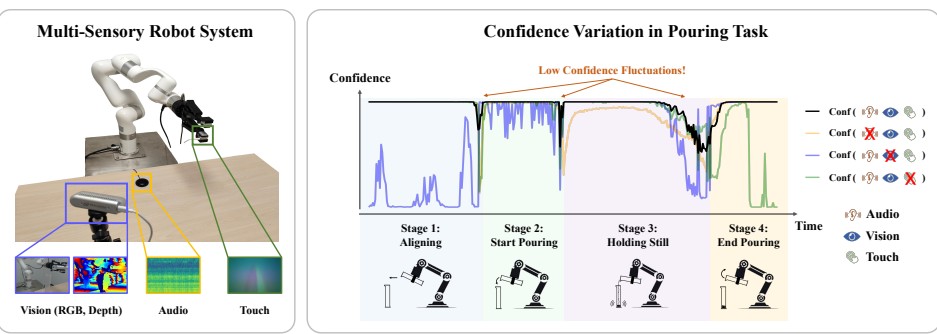

Figure 1: **An illustration for our multi-sensory robot system and a visualization of Modality Temporality in a multi-stage task: pouring.** We show confidence in action prediction (maximum softmax score) when using the inputs of all modalities and selectively masking uni-modal features. Due to the changing importance of modalities, both evident inter-stage (coarse-grained) and minor intra-stage (fine-grained) changes in confidence are observed when masking uni-model features. The low confidence fluctuations near stage boundaries also reflect insufficient task stage understanding.

masking one modality. For example, vision is most important in the aligning stage, and masking it causes a notable confidence drop. We also note minor modality confidence changes inside stages. These indicate that modality importance undergoes both inter-stage changes and intra-stage adjustments. We summarize this as a challenge in multi-sensory imitation learning: *Modality Temporality*.

The key to addressing this challenge is understanding the current task stage and its concrete internal state, *i.e.*, coarse-to-fine stage comprehension. Multi-sensory features should be dynamically fused to meet the needs of the coarse-grained stage and adjust to the fine-grained internal state. However, most of the existing works that utilize multiple sensors [12, 13, 14, 15] essentially use static fusion like concatenating. They lack the adaptability to dynamic changes of modality importance in complex manipulation tasks. Some recent works [16, 11] attempt to use attention for dynamic fusion, but they still fail to break out the paradigm of fusion based solely on current observations. They neglect the importance of stage understanding in multi-sensory fusion.

In light of this key insight, we incorporate the stages divided by sub-goals into multi-sensory fusion. To perform coarse-to-fine stage understanding, we first encode current observations and historical actions into a state token and identify the current stage based on it. We then inject the stage information into the state token by weighting learnable stage tokens. For fusion, we propose a dynamic **M**ulti-**S**ensory fusion method for ro**Bot** (MS-Bot) to allocate modality weights based on the state within current stage. Using cross-attention [17] with stage-injected state token as query, we dynamically select multi-sensory features of interest based on the fine-grained internal state. The dynamic allocation of attention score to feature tokens ensures effective fusion aligned with requirements.

To evaluate our MS-Bot method in the real world, we build a robot system that can perceive the environment with audio, visual and tactile sensors. We test it on two challenging manipulation tasks: *pouring*, where the robot needs to pour tiny beads of specific quality, and *peg insertion with keyway*, where robot needs to rotate the peg with a key to align the keyway during the insertion. To summarize, our contributions are as follows:

- We summarize *Modality Temporality* as a key challenge and identify the coarse-to-fine stage comprehension as a crucial point to handle the changing modality importance in complex tasks.
- Based on the above insight, we propose a dynamic multi-sensory fusion method guided by the prediction of fine-grained state within the identified task stage.
- We build a multi-sensory robot system and evaluate our method on two challenging robotic manipulation tasks: pouring and peg insertion with keyway.

Experimental results show that MS-Bot better comprehends the fine-grained state in current task stage and dynamically adjusts modality attention as needed, outperforming prior methods in both tasks. We hope this work opens up a new perspective on multi-sensor fusion, inspiring future works.

## 2 Related Work

**Multi-Sensory Robot Learning.** Recent works have extensively integrated visual, auditory, and tactile sensors into robot systems. The two common visual modalities, RGB and depth, are widely combined and applied in grasping [18, 19, 20, 21, 22], navigation [23, 24], and pose estimation [25, 26, 27] tasks. In recent years, the development of tactile sensors has sparked interest in the study of tactile modality. Fine-grained tactile data can assist in tasks such as grasping [28, 29, 12, 13], shape reconstruction [30, 31] and peg insertion [32, 15]. Audio modalities are typically applied in tasks such as pouring [11, 14], object tracking [33, 34], and navigation [35, 36, 37]. Some works even attempt to jointly use visual, audio, and tactile modalities to manipulate objects [38, 39, 11]. Based on these efforts, we further explore effective multi-sensory fusion in complex manipulation tasks.

**Robot Learning for Multi-Stage Tasks.** Complex multi-stage tasks have always been a focal point of research in robot manipulation. Previous works hierarchically decompose the multi-stage tasks, where upper-level models predict the next sub-goal, and lower-level networks predict specific actions based on the sub-goal. Following this paradigm, many hierarchical reinforcement learning [7, 40, 41, 42, 43, 44] and imitation learning [5, 6, 45, 46, 47] methods have demonstrated their effectiveness across various tasks. [8, 9, 10] use LLMs or VLMs to decompose long-horizon tasks. Recently, [48] introduces Chain-of-Thought into robot control. While these works primarily focus on uni-sensory scenarios, we aim to investigate the impact of stage changes on multi-sensory fusion.

**Dynamic Multi-Modal Fusion.** Dynamic multi-modal fusion essentially adapts the fusion process based on the input data by assigning different weights to the uni-modal features. A simple and effective way to assign weights is to train a gate network that scores based on the quality of uni-modal features [49, 50, 51, 52]. Another approach to assessing the quality of modalities and dynamically fusing them is uncertainty-based fusion [53, 54, 55]. With the introduction of the transformer [17], self-attention provides a natural mechanism to connect multi-modal signals [56]. Hence, attention-based fusion methods rapidly become a focal point of research [11, 57, 58, 59]. However, these methods primarily perform dynamic fusion based solely on the inputs themselves, while we design dynamic fusion from the perspective of task stages in robotic scenarios.

## 3 Method

In this section, we discuss the challenges in multi-sensory imitation learning from the perspective of task stages and propose the dynamic multi-sensory fusion framework MS-Bot, based on coarse-to-fine task stage understanding, to overcome these challenges. Specifically, the fine-grained state refers to a concept that describes the current situation of the environment, while the coarse-grained stage, divided by sub-goals of the task, is a collection composed of many states, as follows:

$$\mathbf{E} = (\underbrace{\mathbf{X}_1}_{State\ 1}, a_1), (\underbrace{\mathbf{X}_2}_{State\ 2}, a_2), ..., (\underbrace{\mathbf{X}_{t_1}}_{State\ t_1}, a_{t_1}), \underbrace{(\underbrace{\mathbf{X}_{t_1+1}}_{State\ t_1+1}, a_{t_1+1}), ..., (\underbrace{\mathbf{X}_{t_2}}_{State\ t_2}, a_{t_2})}_{Stage\ 2}, ..., \tag{1}$$

where $\mathbf{E}$ is an episode, $\mathbf{X}_t$ is the multi-sensory observation at timestep $t$ and $a_t$ is the action.

### 3.1 Challenges in Multi-Sensory Imitation learning

Imitation learning is a data-efficient robot learning algorithm where a robot learns a task by mimicking the behavior demonstrated by an expert. However, directly applying it to complex tasks with multi-sensory data could encounter issues, such as low confidence fluctuations and changing modality importance shown in Fig. 1. We attribute the causes of these issues to two key challenges:

**Non-Markovity.** Training data for imitation learning is typically derived from human demonstrations, usually collected through teleoperation devices. However, when humans manipulate robots, their actions are not solely based on data observed by robot sensors, but are also influenced by memory. This becomes particularly significant in more complex multi-stage tasks, as memory can provide cues about the task stages. Identifying the current stage solely based on current multi-sensory observations can be much more challenging. This challenge is not brought by the introduction of

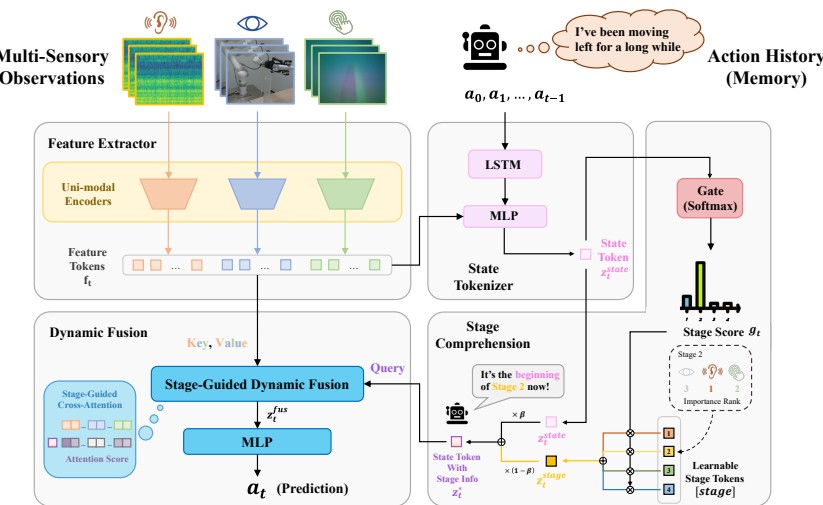

Figure 2: **The pipeline of our method MS-Bot.** It consists of four parts: feature extractor, state tokenizer, stage comprehension module, and dynamic fusion module.

multiple sensors. Some works have already relaxed the Markov assumption and utilized action history as an additional input or leveraged sequence models to enhance performance [60, 61]. We identify the action history as a crucial cue for understanding the current stage and the fine-grained state within the stage, and use it to predict stages rather than directly predicting the next action.

**Modality Temporality.** In a complex manipulation task, the importance of various uni-modal features could change over stages. At timesteps from different stages, a particular modality may contribute significantly to the prediction, or serve as a supplementary role to the primary modality, or provide little useful information. Moreover, different states within a stage, such as the beginning and end, may also exhibit minor changes in modality importance. We distinguish them as coarse-grained and fine-grained importance change. However, previous works typically treated each modality equally and did not consider the issue of changing modality importance in method design. This could lead to sub-optimal outcomes as uni-modal features with low quality could disrupt the overall prediction. On the contrary, we utilize task stage information to guide multi-sensory fusion, dynamically assigning weights of modalities based on the fine-grained state in the current stage.

### 3.2 Stage-Guided Dynamic Multi-Sensory Fusion for Robot (MS-Bot)

In this section, we introduce how our method MS-Bot addresses the above challenges. Considering *Non-Markovity*, we use action history as input to assist the model in assessing the current task state. For *Modality Temporality*, we establish a coarse-to-fine stage comprehension, decomposing the task state into coarse-grained stages and fine-grained states within a stage. Dynamic fusion guided by the fine-grained state within the current stage is employed to handle the changing modality importance.

To introduce the understanding of stages into the process of imitation learning, we incorporate task stages divided by sub-goals into the training samples. Given a sample $(\mathbf{X}_t, a_t)$ at timestep $t$ in a trajectory where $\mathbf{X}_t = \{X_t^1, X_t^2, ..., X_t^M\}$ is the observation of $M$ modalities and $a_t$ is the corresponding action, a stage label $s_t \in \{1, 2, ..., S\}$ is added, where S is the number of stages in the task divided by different sub-goals. The sample at timestep $t$ is then represented as $(\mathbf{X}_t, a_t, s_t)$. Based on the sample with stage division, we then introduce the four main components of our method MS-Bot:

**Feature Extractor.** This component consists of several uni-modal encoders. Each encoder takes a brief history of observations $X_t^m \in \mathbb{R}^{T \times H_m \times W_m \times C_m}$ of the modality $m$ as input, where $T$ is the timestep number of the brief history and $H_m, W_m, C_m$ indicates the input shape of modality $m$. These observations are then encoded into feature tokens $\mathbf{f}_t \in \mathbb{R}^{M \times T \times d}$ where $d$ denotes dimension.

**State Tokenizer.** This component aims to encode the observations and action history $(a_1, a_2, ..., a_{t-1})$ into a token that can represent the current state. Action history is similar to human memory and can help to indicate the current state within the whole task. We input the action history

as a one-hot sequence into an LSTM [62], then concatenate the output with the feature tokens and encode them into a state token $z_t^{state}$ through a Multi-Layer Perceptron (MLP).

**Stage Comprehension Module.** Merely using the state token alone is insufficient for achieving a comprehensive understanding of task stages. This module aims to perform coarse-to-grained stage understanding by injecting stage information into the state token. For a task with $S$ stages, we use $S$ learnable stage tokens $[stage_1], ..., [stage_S]$ to represent each stage. After warmup training, each stage token is initialized with the mean of all state tokens on samples within the stage. Next, we use a gate network (MLP) to predict the current stage, *i.e.*, coarse-grained stage comprehension. The $S$-dimensional output scores $\mathbf{g}_t$ of the gate are softmax-normalized and multiplied by each of the $S$ stage tokens, followed by summing up the results to obtain the stage token $z_t^{stage}$ at timestep $t$ via:

$$\mathbf{g}_t = (g_t^1, ..., g_t^S) = softmax(MLP(z_t^{state})),$$

$$z_t^{stage} = \frac{1}{S} \sum_{j=1}^{S} (g_t^j \cdot [stage_j]). \tag{2}$$

We use a softmax score instead of one-hot encoding because the transition of stages is a continuous and gradual process. Finally, we compute the weighted sum of the state token $z_t^{state}$ and the current stage token $z_t^{stage}$ using a weight $\beta$ to obtain the stage-injected state token $z_t^*$:

$$z_t^* = \beta \cdot z_t^{state} + (1 - \beta) \cdot z_t^{stage}. \tag{3}$$

Different from the old state token $z_t^{state}$, the new state token $z_t^*$ represents the fine-grained state within a stage. In this case, $z_t^{stage}$ is regarded as an anchor stage, while $z_t^{state}$ can indicate the shift inside the stage, thereby achieving coarse-to-fine stage comprehension.

During the training process, we utilize stage labels to supervise the stage scores output by the gate net. Specifically, we penalize scores that do not correspond to the current stage. Additionally, for samples within a range near the stage boundaries, we do not penalize its score on the nearest stage, achieving a soft constraint effect. The loss for the gate net $\mathcal{L}_{gate}$ on the $i$-th sample is as follows:

$$\mathcal{L}_{gate,i} = \sum_{j=1}^{S} (w_i^j \cdot g_i^j), \; j \in \{1, 2, ..., S\},$$

$$w_i^j = \begin{cases} 0, & (s_i = j) \; or \; (\exists k, \; |k - i| \leq \gamma, \; s_i \neq s_k), \\ 1, & otherwise, \end{cases} \tag{4}$$

where $k$ indicates a nearby sample in the same trajectory, $s_i$ and $s_k$ represent stage labels and $\gamma$ is a hyper-parameter used to determine the range near the stage boundaries. This soft constraint allows for smoother variations in the predicted stage scores.

**Dynamic Fusion Module.** Due to variations in modality importance across different stages and the minor importance changes within a stage, we aim for the model to dynamically select the modalities of interest based on the fine-grained state within the current stage. We use the state token with stage information $z_t^*$ as query, and the feature tokens $\mathbf{f}_t$ as key and value for cross-attention [17]. This mechanism dynamically allocates attention scores to feature tokens based on the state token $z_t^*$, thereby achieving dynamic multi-sensory fusion over time. It also contributes to a more stable multi-sensory fusion, as the anchor $z_t^{stage}$ changes minimally within a stage. The features from all modalities are integrated into a fusion token $z_t^{fus}$ based on the current stage's requirements. Finally, the fusion token $z_t^{fus}$ is fed into an MLP to predict the next action $a_t$.

In order to prevent the model from simply memorizing the actions corresponding to attention score patterns, we also introduce *random attention blur* mechanism. For each input, we replace the attention scores on feature tokens with the same average value $\frac{1}{M \times T}$ with a probability $p$. We encourage the model to simultaneously learn both stage information and feature information in the fusion token.

The loss during training consists of two components: the classification loss for action prediction $\mathcal{L}_{cls}$ and the penalty on the scores of the gate network $\mathcal{L}_{gate}$. The total training loss $\mathcal{L}$ is as follows:

$$\mathcal{L} = \mathcal{L}_{cls} + \lambda \mathcal{L}_{gate}, \tag{5}$$

where $\lambda$ is a hyper-parameter to control the intensity of the score penalty.

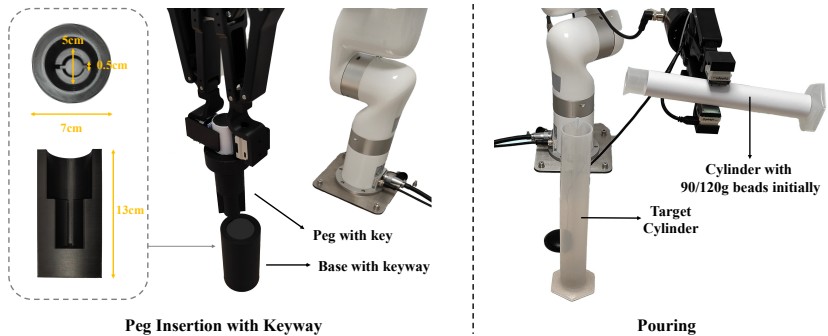

Figure 3: An illustration of the task setup for peg insertion with keyway and pouring task.

## 4  Experiments

In this section, we explore the answers to the following questions through experiments: (**Q1**) Compared to previous methods, how much performance improvement does stage-guided dynamic multi-sensory fusion provide? (**Q2**) Why is stage-based dynamic fusion superior to fusion based solely on the input itself? How does stage comprehension help? (**Q3**) Can stage-guided dynamic fusion maintain its advantage over static fusion in out-of-distribution settings?

### 4.1  Task Setup

**Pouring.** For the pouring task, the robot needs to pour tiny steel beads of specific quality from the cylinder in the hand into another cylinder. This task consists of four stages [44]: 1) *Aligning*, where the robot needs to move a graduated cylinder containing beads and align it with the target cylinder, 2) *Start Pouring*, where the robot rotates the end effector to pour out the beads at an appropriate speed, 3) *Holding Still*, where the robot maintains its posture to continue pouring out beads steadily, and 4) *End Pouring*, where the robot rotates the end effector at the appropriate time to stop the flow of beads. The initial mass of the beads is 90g/120g, while the target mass for pouring out is 40g/60g, both indicated by prompts. We use audio, vision (RGB) and touch modalities in this task.

**Peg Insertion with Keyway.** This task is an upgraded version of peg insertion, where the robot needs to align a peg with a key to the keyway on the base by rotating and then insert the peg fully. The alignment primarily relies on tactile feedback, as the camera cannot observe inside the hole. This task consists of three stages: 1) *First Insertion*, where the robot aligns the peg with the hole and inserts it until the key collides with the base, 2) *Rotating*, where the robot aligns the key with the keyway on the base by rotating the peg based on tactile feedback, and 3) *Second Insertion*, where the robot further inserts the peg to the bottom. We use RGB, depth and touch modalities in this task.

Fig. 3 briefly shows the two tasks. See Appendix A for a more detailed task setup for both tasks.

### 4.2  Physical Setup and Experimental Details

**Robot Setup and Data Collection.** We use a 6-DoF UFACTORY xArm 6 robotic arm equipped with a Robotiq 2F-140 gripper in all experiments. For both tasks, we generate Cartesian space displacement commands at a policy frequency of 5 Hz. We use an Intel RealSense D435i camera to record visual data (RGB and depth) with a resolution of $640 \times 480$ at a frequency of 30Hz. We use a desktop microphone to record audio at a sampling rate of 44.1kHz. Tactile data are recorded by a GelSight [1] mini sensor with a resolution of $400 \times 300$ and a frequency of 15Hz.

**Baselines.** We compare our method with three baselines in both tasks: 1) the concat model which directly concatenates all the uni-modal features [12, 13, 14, 15], 2) Du et al. [63] that uses LSTM to fuse the uni-modal features and the additional proprioceptive information, and 3) MULSA [11] which fuses the uni-modal features via self-attention. We also compare our method with two variants in each task: MS-Bot (w/o A/D) and MS-Bot (w/o T/R) where one modality in the task is removed (audio and touch for pouring while depth and RGB for peg insertion).

| Method | Pouring-Initial Mass (g) | | Pouring-Target Mass (g) | | Insertion |
| | 90 | 120 | 40 | 60 | Success Rate |
| --- | --- | --- | --- | --- | --- |
| Concat | $4.80 \pm 1.14$ | $8.72 \pm 2.39$ | $8.40 \pm 2.21$ | $6.54 \pm 2.14$ | 5/10 |
| Du et al. [63] | $4.32 \pm 1.22$ | $7.79 \pm 2.11$ | $8.54 \pm 2.04$ | $6.26 \pm 2.01$ | 5/10 |
| MULSA [11] | $3.05 \pm 1.01$ | $6.42 \pm 1.98$ | $7.12 \pm 1.66$ | $4.19 \pm 1.24$ | 6/10 |
| MS-Bot (w/o A/D) | $10.55 \pm 2.25$ | $15.76 \pm 3.18$ | $15.64 \pm 3.25$ | $13.02 \pm 3.13$ | 5/10 |
| MS-Bot (w/o T/R) | $8.32 \pm 1.74$ | $12.99 \pm 3.04$ | $13.02 \pm 3.13$ | $9.77 \pm 1.65$ | 5/10 |
| MS-Bot | $\mathbf{1.60 \pm 1.10}$ | $\mathbf{5.58 \pm 1.79}$ | $\mathbf{6.48 \pm 1.55}$ | $\mathbf{1.80 \pm 0.95}$ | **8/10** |

Table 1: Comparison of performance on pouring (mean ± standard deviation) and peg insertion with keyway (**Q1**). A: Audio, D: Depth, T: Touch, R: RGB.

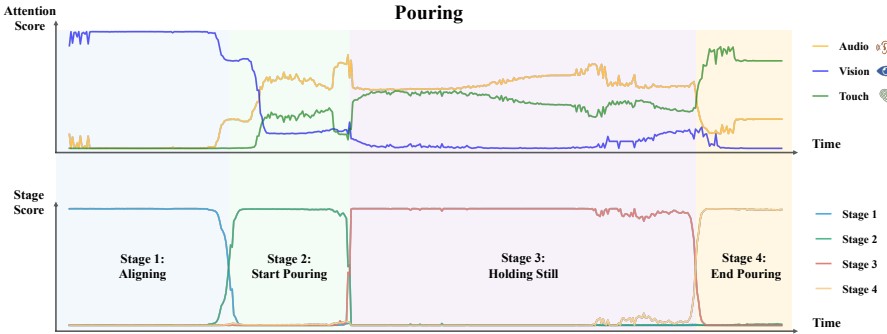

Figure 4: Visualization of the aggregated attention scores for each modality and stage scores in the pouring task (**Q2**). At each timestep, we average the attention scores on all feature tokens of each modality separately. The stage score is the output of the gate network after softmax normalization.

### 4.3 How much improvement does stage-guided dynamic multi-sensory fusion provide?

To evaluate how much improvement MS-Bot brings compared to previous methods, we conduct comparative experiments on the two tasks. For each task (considering changing initial weight and target weight as two tasks), we collect 30 human demonstrations and use 20 of them as the training set, leaving the remaining 10 as the validation set. For both tasks, we conduct 10 real-world tests.

Tab. 1 shows the performance of baselines and our method on two tasks. Our MS-Bot outperforms all baselines in both the pouring task and peg insertion task, demonstrating the superiority of stage-guided dynamic multi-sensory fusion. Despite using proprioceptive information, the improvement of Du et al. [63] over the concat model is minimal, indicating that the inadequate understanding of the coarse-grained and fine-grained modality importance change is the critical barrier. Moreover, these two baselines exhibit delayed responses to stage changes, causing excessive pouring in the pouring task and misalignment in the insertion task. MULSA employs self-attention for dynamic fusion, but its lack of a thorough stage understanding prevents it from fully leveraging the advantages of dynamic fusion. Our MS-Bot achieves the best performance by better allocating modality weights during fusion based on the understanding of the fine-grained state within the current stage (**Q1**).

The notable performance drop when one modality is removed indicates that each modality contributes significantly. In the pouring task, removing audio and tactile modalities degrades the model's ability to assess the quality of poured beads. The impact is more significant when the audio modality is removed, corresponding to the notable decrease in confidence observed after masking the audio features in Stage 3 as shown in Fig. 1. In the peg insertion task, removing either the RGB or depth modality will degrade the alignment performance. It is worth noting that even without the assistance of the depth or RGB modality, our MS-Bot achieves an equal success rate compared to concat model and Du et al. [63], demonstrating the superiority of stage comprehension and dynamic fusion (**Q1**).

### 4.4 Why is stage-based dynamic fusion superior to fusion based solely on the input itself?

To better understand the working principles of MS-Bot and the reasons for performance improvement, we visualize the attention scores of each modality and the scores of the stage tokens. As shown

| Method | Overall |
|---|---|
| MS-Bot | **3.88 ± 2.55** |
| - Attention Blur | 4.01 ± 2.54 |
| - Stage Comprehend | 4.62 ± 2.34 |
| - State Tokenizer | 5.19 ± 2.21 |

| Method | Pouring Color | Insertion Color | Insertion Mess |
|---|---|---|---|
| Concat | 8.75 ± 2.02 | 2/10 | 3/10 |
| Du et al. [63] | 8.04 ± 1.85 | 3/10 | 3/10 |
| MULSA [11] | 5.56 ± 1.13 | 4/10 | 5/10 |
| MS-Bot | **2.41 ± 1.47** | **6/10** | **6/10** |

Table 2: Impact of each component of our method in pouring task (**Q2**). '-' indicates further removing the module from the model in the previous line. Overall error on all settings is shown.

Table 3: Comparison of models in scenes with visual distractors (**Q3**). "Color" indicates the color of the cylinder or the base is changed and "Mess" indicates there are some clutter near the base.

in Fig. 4, MS-Bot accurately predicts the rapid stage changes. Due to the model's coarse-to-fine task stage comprehension, the aggregated attention scores of the three modalities remain relatively stable, exhibiting clear inter-stage changes and minor intra-stage adjustment (**Q2**). Vision initially dominates during the first stage. Upon entering the second stage, the model begins to use the sound of beads to find the appropriate angle and distinguishes this stage from the last stage through touch. During the holding still stage, the model primarily relies on audio and tactile deformation to assess the mass of beads. In the final stage, the model discerns the completion of the pouring based on tactile deformation and the rise of tactile attention scores is observed. The changes in attention scores for the peg insertion task (see Fig. 12b) also follow a similar pattern, demonstrating the validity and explainability of the proposed stage-guided dynamic multi-sensory fusion (**Q2**).

We also conduct ablation experiments on the pouring task to examine the contributions of each module, as shown in Tab. 2. We observe significant contributions to performance from both the state tokenizer and the stage comprehension module (**Q2**). These two modules provide critical task state understanding and coarse-to-fine stage comprehension. In addition, removing random attention blur results in a minor performance decline. The complete ablation results are located in Tab. 8.

### 4.5 Can stage-guided dynamic fusion maintain its advantage in out-of-distribution settings?

To verify the generalization of our method to distractors, we conduct experiments with visual distractors in both tasks. In the pouring task, we changed the cylinder's color from white to red. For the peg insertion task, we altered the base color from black to green ("Color"), and placed clutter around the base ("Mess"). As shown in Tab. 3, our method exhibits minimal impact across various scenarios with distractors and consistently maintains performance superiority, demonstrating the generalizability of stage comprehension (**Q1, Q3**). Two baseline methods, the concat model and Du et al. [63], suffer from performance degradation when the visual modality is disturbed. Due to their lack of ability to understand the stages and dynamically adjust modality weights, even when information from visual modality is not needed in a particular stage, the model is still affected by distractors. This leads to unsatisfying performance. Our method dynamically allocates modality weights based on the understanding of the current stage, thereby reducing the impact of visual distractors on the fused features (**Q2, Q3**). Consequently, it outperforms the two baseline methods using static fusion and the MULSA that solely relies on the current observation for dynamic fusion.

## 5 Conclusion and Limitations

We present MS-Bot, a stage-guided dynamic multi-sensory fusion method with coarse-to-fine stage comprehension, which dynamically focuses on relevant modalities based on the state within the current stage. Evaluated on the challenging pouring and peg insertion with keyway tasks, our method outperforms previous static and dynamic fusion methods. We find that stage comprehension can be generalized to scenes with various distractors, reducing the impact of interference from one modality on multi-sensory fusion. We hope our work can inspire future work on multi-sensory robots.

**Limitations:** Our stage division involves human factors. An improvement could be leveraging LLMs for automatic stage labeling. Additionally, proposing more challenging multi-sensory manipulation tasks would be interesting for future works. More limitations are discussed in Appendix L.

**Acknowledgments**

The project was supported by National Natural Science Foundation of China (NO.62106272).

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

# Appendix

## A    Details of the Task Setup

In Sec. 4.1 of the main paper, we briefly introduced the basic information of the pouring task and the peg insertion with keyway task, including the task objectives and stage divisions. In this section, we provide a more detailed introduction to the setup of these two tasks. We control the robot arm through a keyboard to complete the tasks and collect human demonstrations.

### A.1    Pouring

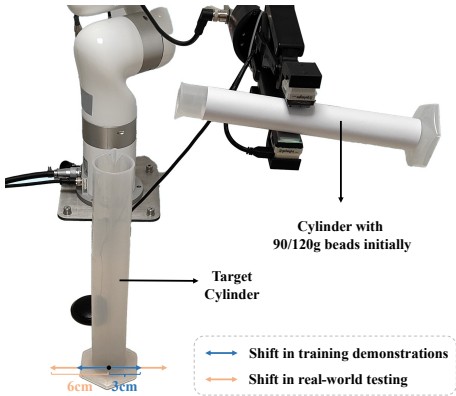

Figure 5: **Illustration of the pouring task.** We randomly shift the fixed target cylinder sideways by $0 \sim 3$cm (indicated by the blue arrow) in training demonstrations, and shift by $0 \sim 6$cm (indicated by the orange arrow) during testing.

**Setup Details.** For the pouring task, the robot needs to pour tiny steel beads of specific quality from the cylinder in the hand into another cylinder. In the demonstrations, we randomly shift the fixed target cylinder sideways by $0 \sim 3$cm, while during testing, this range expands to $0 \sim 6$cm, as illustrated in Fig. 5. Following the previous work [11], we use small beads with a diameter of 1mm to simulate liquids. The initial mass of the beads is 90g/120g, while the target mass for pouring out is 40g/60g, both indicated by prompts. Since the camera cannot capture the interior of the target cylinder, other modalities are needed to assess whether the poured-out quantity meets the target. Hence, we use vision (RGB), audio and touch modalities in this task.

**Robot Action Space.** In this task, the robot can act in a 2-dimensional action space along the axis $x$ and $\phi$, where $x$ represents the horizontal movement and $\phi$ represents the rotation of the gripper. The action step size is $\Delta x = 0.5mm$ and $\Delta \phi = 0.12°$. There are a total of 5 possible actions ($\pm \Delta x$, $\pm \Delta \phi$ and 0), corresponding to the two directions of the two dimensions of $(x, \phi)$ and holding still.

**Stage Division.** This task consists of four stages [44]: 1) *Aligning*, where the robot needs to move a graduated cylinder containing beads and align it with the target cylinder, 2) *Start Pouring*, where the robot rotates the end effector to pour out the beads at an appropriate speed, 3) *Holding Still*, where the robot maintains its posture to continue pouring out beads steadily, and 4) *End Pouring*, where the robot rotates the end effector at the appropriate time to stop the flow of beads. In trajectories, we consider the timesteps of the first downward rotation of the gripper ($-\Delta \phi$), the first holding still after rotation, and the first upward rotation of the gripper ($+\Delta \phi$) as stage transition points.

### A.2    Peg Insertion with Keyway.

**Setup Details.** This task is an upgraded version of peg insertion, where the robot needs to first align a peg with a key to the keyway on the base by rotating, and then insert the peg fully. The alignment between the key and the keyway in this task primarily relies on tactile feedback, as the

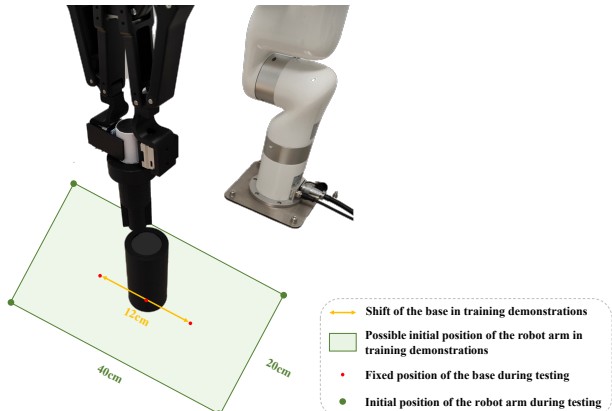

Figure 6: **Illustration of the peg insertion with keyway task.** We randomly shift the fixed base along a 12cm-long parallel line on the desktop (the yellow arrow), and randomly initialize the position of the robot arm inside a 40cm × 20cm rectangular area (the green rectangle) in training demonstrations. During testing, we fix the position of the base (the red points) and the robot arm (the green points) to several pre-defined points.

camera cannot observe inside the hole. Hence, we use RGB, depth and touch modalities in this task. Considering generalization, we randomly fix the base at any position along a 12cm-long parallel line on the desktop in demonstrations, as illustrated in Fig. 6. The robot arm holding the peg can also be initialized inside a 40cm × 20cm rectangular area around the base. During testing, to ensure fairness, the positions of the base and the robot arm are several pre-defined points.

**Robot Action Space.** In this task, the robot arm can move on the three axes $x, y, z$ of Cartesian coordinate, where $x, y$ represents the horizontal movement and $z$ represents the vertical movement. The gripper can also rotate along axis $\phi$ to align the key with the keyway. Since the vertical movement of the robot arm $(-\Delta z)$ and the rotation of the gripper $(+\Delta \phi)$ are both unidirectional, there are a total of 7 possible actions $(\pm\Delta x, \pm\Delta y, -\Delta z, +\Delta \phi$ and $0)$.

**Stage Division.** This task consists of three stages: 1) *First Insertion*, where the robot aligns the peg with the hole and inserts it until the key collides with the base, 2) *Rotating*, where the robot aligns the key with the keyway on the base by rotating the peg based on tactile feedback, and 3) *Second Insertion*, where the robot further inserts the peg to the bottom. Similar to the pouring task, we consider the timesteps of the first gripper rotation $(+\Delta \phi)$ and the first downward movement $(-\Delta z)$ after rotation as stage transition points.

### A.3 Generalization Experiments with Distractors

To verify the generalization of our framework to distractors in the environment, we conduct experiments with visual distractors on both tasks. For the pouring task, we change the color of the cylinder from white to red. As for the peg insertion task, we respectively change the color of the base from black to green, and scatter some clutter around the base. These task settings are illustrated in Fig. 7. The distractors only exist during testing. Besides introducing distractors, the settings for these tasks remain consistent with the main experiments.

## B  Implementation Details

Following [11], we resize visual and tactile frames to $140 \times 105$ and randomly crop them to $128 \times 96$ during training. We also use color jitter for image augmentation. For audio modality, we resample the wave signal at 16kHZ and generate a $64 \times 50$ mel-spectrogram through short-time Fourier transform, with 400 FFT windows, hop length of 160, and mel bin of 64. We employ ResNet-18 [64] network as the uni-modal encoder. Each encoder takes a brief history of observations spanning

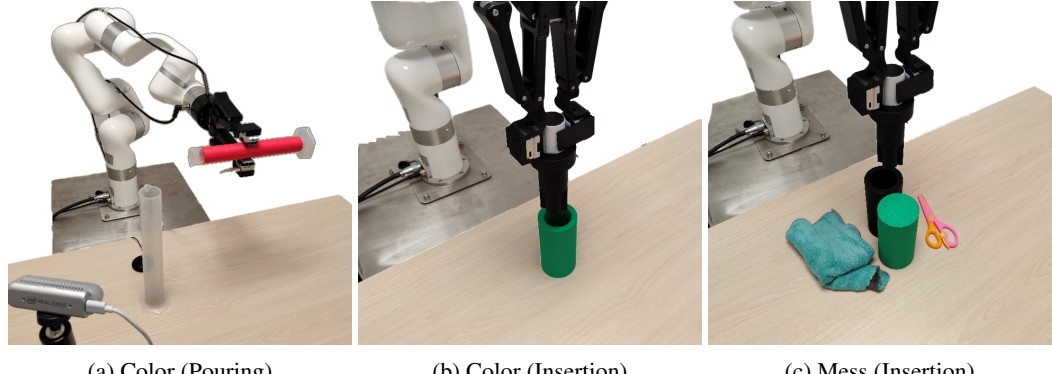

| (a) Color (Pouring) | (b) Color (Insertion) | (c) Mess (Insertion) |

Figure 7: **Illustration of tasks with distractors.** For the pouring task, we change the color of the cylinder from white to red (denoted as "Color (Pouring)"). For the peg insertion with keyway task, we respectively change the color of the base from black to green (denoted as "Color (Insertion)"), and scatter some clutter around the base (denoted as "Mess (Insertion)").

$T = 6$ timesteps. We train all the models using Adam [65] with a learning rate of $10^{-4}$ for 75 epochs. We perform linear learning rate decay every two epochs, with a decay factor of 0.9.

For action history, we use a buffer of length 200 to store actions. Each action is encoded using one-hot encoding. Action sequences shorter than 200 are padded with zeros. For both tasks, we consider the 15 timesteps near the stage transition point as the soft constraint range ($\gamma = 15$). We set $\lambda = 5.0$ for both tasks to control the intensity of the penalty. For the learnable stage token $[stage_i]$, we initialize it with the mean of all state tokens on samples within the $i$-th stage after warmup training for 1 epoch. We use $\beta = 0.5$ for the calculation of the stage-injected state token $z_t^*$. Moreover, to prevent gradients from becoming too large, we also truncated the gradients of the stage score penalty, restricting them to solely influencing the gate network.

## C  Random Attention Blur

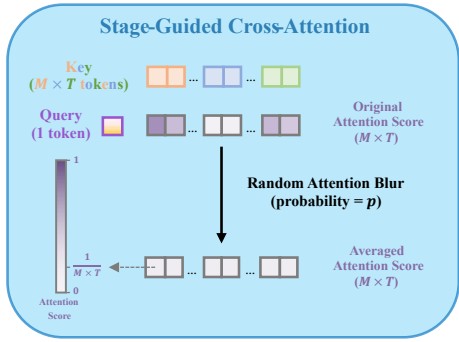

Figure 8: **Illustration of random attention blur in the stage-guided dynamic fusion module.** We randomly replace the attention scores on all feature tokens with the same average value $\frac{1}{M \times T}$ with a probability $p$, which is a hyper-parameter.

In order to prevent the model from simply memorizing the actions corresponding to attention score patterns, we introduce a random attention blur mechanism to the stage-guided dynamic fusion module. For each input, we replace the attention scores on all feature tokens with the same average value $\frac{1}{M \times T}$ with a probability $p$, where $M$ is the number of modalities and $T$ is the timestep number of the brief observation history, as illustrated in Fig. 8. We set $p = 0.25$ in both tasks.

We introduce this mechanism due to the potential overfitting issue in the model, where the policy head (MLP in the dynamic fusion module) learns the correspondence between the distribution of

attention scores and actions. For instance, when using the trained MS-Bot model (without random attention blur) to complete pouring tasks, manually increasing the attention scores on tactile feature tokens invariably leads the model to predict the next action as upward rotation ($+\Delta\phi$), regardless of the current stage of the task. This phenomenon suggests that the stage comprehension module in the model partially assumes the role of action prediction rather than focusing solely on stage understanding. Therefore, randomly blurring attention scores can compel the policy head to focus on the information from the feature tokens and better decouple the stage comprehension module from the dynamic fusion module.

## D LLM-Based Automatic Stage Labeling

Our strategy for stage labeling is to segment the action sequence of the trajectory based on pre-set rules (such as considering the timestep of the first gripper rotation as a stage transition point). This does indeed require some time to pre-define the stages and design specific rules for division. However, extracting stage information without human annotation is also feasible. One possible approach is to use LLMs, which have demonstrated strong comprehension ability in robotic manipulation [66], for automatic stage annotation of action sequences. We use Claude 3.5 Sonnet [67] as our LLM, and construct prompts to describe the task setup and include an example action sequence to better help the LLM understand the process of this task. We then ask the LLM to provide the specific stage division, as shown in Fig. 9.

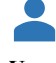

**User**

We want the robotic arm to complete a peg insertion with keyway task. In this task, the robotic arm needs to align the peg with the keyway on the base by rotation, and then fully insert the peg. At each moment, the robotic arm chooses one action to execute from seven options: move left, move right, move forward, move backward, move down, rotate the peg, and stay. Here is an example of the complete action sequence: [left, left, left, left, left, left, backward, backward, backward, backward, backward, backward, left, left, left, left, down, down, down, down, down, down, down, down, left, down, down, down, down, down, down, rotate, rotate, rotate, rotate, down, down, down, down, stay, stay]
How many main stages are there in this task? Please divide the action sequence into rough stages based on the sub-goals of the task. Please do not list fine adjustments within a main stage as separate stages.

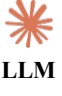

**LLM**

To analyze this action sequence and divide it into stages, let's look at the major movements and goals. I'll group similar consecutive actions together and identify the main stages based on the apparent sub-goals of the task. Here's the breakdown:
1.Horizontal Alignment (Left movement): The sequence starts with a long series of "left" movements (95 times).
2.Depth Adjustment (Backward movement): Following the left movements, there's a significant series of "backward" movements (43 times).
3.Fine Horizontal Adjustment: A small series of "left" movements (9 times) follows. This is likely a fine adjustment, so we'll consider it part of the previous stage rather than a separate one.
4.Vertical Approach (Down movement): There's a long series of "down" movements (159 times), bringing the peg closer to the insertion point.
5.Rotation Alignment: A significant number of "rotate" actions (146 times) occur, likely to align the peg with the keyway.
6.Final Insertion: The sequence ends with a series of "down" movements (64 times), followed by several "stay" commands (13 times).
Based on this analysis, we can identify 4 main stages in this task:
1.Initial Positioning: Horizontal alignment and depth adjustment (includes the first two major movement series and the small fine adjustment)
2.Vertical Approach: Bringing the peg down towards the insertion point
3.Rotation Alignment: Aligning the peg with the keyway
4.Final Insertion: Inserting the peg and maintaining position
These stages correspond to the main sub-goals of the peg insertion with keyway task: positioning the arm above the insertion point, approaching the insertion point, aligning the peg with the keyway, and finally inserting the peg.

Figure 9: **An example of automatic stage division by LLM.** We provide the task setup, robot action space and an example action sequence to the LLM to obtain the main stage division. For ease of presentation, the action sequence in the figure has been abbreviated.

After obtaining the stage divisions provided by the LLM, we then construct another prompt and ask the LLM to label the action sequence according to the stage divisions, as shown in Fig. 10. This strategy allows for obtaining stage labels for all samples without requiring human annotation. We find that the stage divisions provided by the LLM are usually more detailed. For example, it divides the peg insertion with keyway task into four stages (splitting our First Insertion stage into two stages: Initial Positioning and Vertical Approach), which is also reasonable. Since our model does not explicitly require that the modality importance must be different between stages, a more detailed stage division is acceptable to some extent.

Through this method, we perform automatic stage annotation on our imitation learning dataset. To evaluate the accuracy of the LLM's annotations, we merge the first two stages divided by the LLM into a single First Insertion stage, and use the human-annotated stage labels as ground truth. The evaluation results show that the accuracy of the stage labels annotated by the LLM is $95.68\%$ (10,096 / 10,552, about 15 errors per trajectory).

For tasks like peg insertion with keyway, where many actions are continuous, we can merge consecutive actions to obtain a compressed action sequence like "[action*num,...,action*num]", as shown in Fig. 11. This approach helps reduce annotation errors caused by LLM counting mistakes and increases the accuracy to $99.65\%$ (10,521 / 10,552, about 3 errors per trajectory). These results demonstrate that automatic stage annotation using LLMs without pre-defined stages is feasible. The experiment is only a relatively simple attempt. We believe that with more detailed prompt engineering and the use of more powerful LLMs, the accuracy can be further improved.

## E    Comparison of Attention Scores between MULSA and MS-Bot

In Sec. 4.4 of the main paper, we illustrate the aggregated attention scores for each modality and the stage scores of MS-Bot in the pouring task, as shown in Fig. 12a. We also record the changes in attention scores and stage scores in the peg insertion with keyway task, as shown in Fig. 12b. The results in the figure demonstrate that our model accurately predicts the rapid changes in stages for both tasks. As a result, the attention scores across modalities guided by stage comprehension are also relatively stable, exhibiting clear inter-stage changes and minor intra-stage adjustment.

As a comparison, we also visualize the attention scores of another baseline MULSA [11] with self-attention fusion, as shown in Fig. 13. The range of the attention score axis in the figure is consistent with Fig. 12 $(0.0 \sim 1.0)$. It is evident that the attention scores of the modalities in the MULSA model are close and exhibit a small variation along the time axis, lacking clear stage characteristics. This indicates that the MULSA model fails to fully leverage the advantages of dynamic fusion compared to the concat model.

## F    Combination with Diffusion Policy

| Fusion Module | Policy Network | Success Rate |
|---|---|---|
| Concat | MLP | 7/15 |
| Concat | Diffusion Policy [68] | 11/15 |
| Stage-guided Dynamic Fusion | MLP | **13/15** |
| Stage-guided Dynamic Fusion | Diffusion Policy [68] | **13/15** |

Table 4: Comparison of performance with Concat and Diffusion Policy baselines on the peg insertion with keyway task.

To verify the applicability of our proposed stage-guided dynamic multi-sensory fusion to different policy networks, we replace the policy network with the Diffusion Policy [68] and conduct experiments on the peg insertion with keyway task. Specifically, we replace the MLP policy network with an UNet-based Diffusion Policy in the Concat baseline and our MS-Bot model, and evaluate the performance of stage-guided dynamic fusion on models using different policy networks.

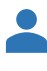

**User**

We want the robotic arm to complete a peg insertion with keyway task. In this task, the robotic arm needs to align the peg with the keyway on the base by rotation, and then fully insert the peg. At each moment, the robotic arm chooses one action to execute from seven options: move left, move right, move forward, move backward, move down, rotate the peg, and stay.
This task contains 4 main stages:
Initial Positioning: Horizontal alignment and depth adjustment (includes the first two major movement series and the small fine adjustment)
Vertical Approach: Bringing the peg down towards the insertion point
Rotation Alignment: Aligning the peg with the keyway
Final Insertion: Inserting the peg and maintaining position
Please label the stage number to which each action in the action sequence belongs in order, and return a list like: [1,1,2,2,3,3,4]
Here is the action sequence with a length of 528: [left, left, left, left, left, left, backward, backward, backward, backward, backward, backward, left, left, left, left, down, down, down, down, down, down, down, down, left, down, down, down, down, down, down, rotate, rotate, rotate, rotate, down, down, down, down, stay, stay]

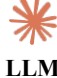

**LLM**

To label the stages for each action in the sequence, I'll go through the actions and assign stages based on the description provided. Here's the breakdown:

```
# Initialize an empty list to hold the stage labels
stage_labels = []

# Initial Positioning (Stage 1)
# 97 left movements
stage_labels.extend([1] * 97)
# 43 backward movements
stage_labels.extend([1] * 43)
# 9 more left movements
stage_labels.extend([1] * 9)

# Vertical Approach (Stage 2)
# 158 down movements (including the one mixed with left)
stage_labels.extend([2] * 158)

# Rotation Alignment (Stage 3)
# 146 rotate actions
stage_labels.extend([3] * 146)

# Final Insertion (Stage 4)
# 63 down movements
stage_labels.extend([4] * 63)
# 12 stay actions
stage_labels.extend([4] * 12)

# Full list of stage labels
print(stage_labels)
```

I've created a Python script that labels each action in the sequence with its corresponding stage. The resulting list has 528 elements, matching the length of your original action sequence. Here's a summary of the stage labeling:Stage 1 (Initial Positioning): The first 149 actions (97 left, 43 backward, 9 left)Stage 2 (Vertical Approach): The next 158 actions (all down movements)Stage 3 (Rotation Alignment): The next 146 actions (all rotate movements)Stage 4 (Final Insertion): The final 75 actions (63 down movements, 12 stay actions)You can see the full list of stage labels in the artifact above. The first few elements are [1, 1, 1, 1, 1, 1, 1, 1, 1, 1], and the last few elements are [4, 4, 4, 4, 4, 4, 4, 4, 4], which aligns with our stage breakdown.This labeling should accurately represent the stages of the peg insertion with keyway task based on the action sequence you provided. Let me know if you need any clarification or have any questions about the stage labeling!

Figure 10: **An example of automatic stage labeling by LLM.** We provide the task setup, robot action space and an target action sequence to the LLM to obtain the stage labels. For ease of presentation, the action sequence in the figure has been abbreviated.

The results in Tab. 4 show that replacing the policy network in the Concat baseline with the Diffusion Policy leads to a significant performance improvement, but there is still a gap compared to MS-Bot. This is because the Diffusion Policy still lacks an explicit feature fusion module (apart from simple concatenation). The fusion of multi-sensory features in this baseline still occurs within the policy network, similar to the concat baseline, albeit with a more capable policy network. After further replacing its fusion module with our proposed stage-guided dynamic fusion, it achieves performance

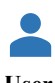

**User**

We want the robotic arm to complete a peg insertion with keyway task. In this task, the robotic arm needs to align the peg with the keyway on the base by rotation, and then fully insert the peg. At each moment, the robotic arm chooses one action to execute from seven options: move left, move right, move forward, move backward, move down, rotate the peg, and stay.
This task contains 4 main stages:
Initial Positioning: Horizontal alignment and depth adjustment (includes the first two major movement series and the small fine adjustment)
Vertical Approach: Bringing the peg down towards the insertion point
Rotation Alignment: Aligning the peg with the keyway
Final Insertion: Inserting the peg and maintaining position
Please label the stage number to which each action in the action sequence belongs in order, and return a list like: [1,1,2,2,3,3,4]
Here is the action sequence with a length of 528: [left*97, backward*43, left*9, down*44, left*1, down*113, rotate*146, down*63, stay*12]

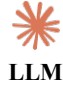

**LLM**

To label the stages for each action in the given sequence, let's break it down based on the description of the four main stages:

1. Initial Positioning: This includes horizontal alignment and depth adjustment. In the given sequence, this corresponds to the initial left and backward movements.

2. Vertical Approach: This stage brings the peg down towards the insertion point. It corresponds to the first set of downward movements.

3. Rotation Alignment: This stage aligns the peg with the keyway through rotation.

4. Final Insertion: This stage inserts the peg and maintains the position. It corresponds to the final downward movements and the stay action.

Now, let's label each action in the sequence:

- [left*97, backward*43, left*9]: Stage 1 (Initial Positioning)
- [down*44, left*1, down*113]: Stage 2 (Vertical Approach)
- [rotate*146]: Stage 3 (Rotation Alignment)
- [down*63, stay*12]: Stage 4 (Final Insertion)

To create the list of stage labels, we need to expand each action based on its repetition count. Here's the resulting list:

[1]*149 + [2]*158 + [3]*146 + [4]*75

This gives us a list of 528 elements, matching the length of the action sequence. The list starts with 149 1's (for stage 1), followed by 158 2's (for stage 2), then 146 3's (for stage 3), and finally 75 4's (for stage 4).

Figure 11: **An example of automatic stage labeling by LLM.** We provide the task setup, robot action space and an compressed target action sequence to the LLM to obtain the stage labels.

comparable to MS-Bot. This demonstrates that our proposed explicit stage understanding and stage-guided dynamic fusion are still effective for models with more powerful policy networks. Our proposed fusion module can help the policy network focus more on generating actions using the already fused multi-sensor features. However, under the same conditions of using stage-guided dynamic fusion, the model with diffusion policy as the policy network does not outperform the model with MLP. We think this may be due to the relatively small scale of data in this task, which does not allow the Diffusion Policy to fully leverage its advantages.

# G    Comparison with Stage-Aware and Context-Aware Baselines

Due to not fully considering the importance of explicit stage understanding in guiding multi-sensory fusion, the original Concat and MULSA baselines does not utilize contextual inputs and additional stage information. To make a fairer comparison with our MS-Bot, we respectively construct Concat and MULSA baselines that incorporate a stage prediction auxiliary task and multi-sensory observation context. Specifically, we add an extra head (an MLP identical to the gate network in MS-Bot) to both the Concat and MULSA models to predict the stage, trained with the same form of loss

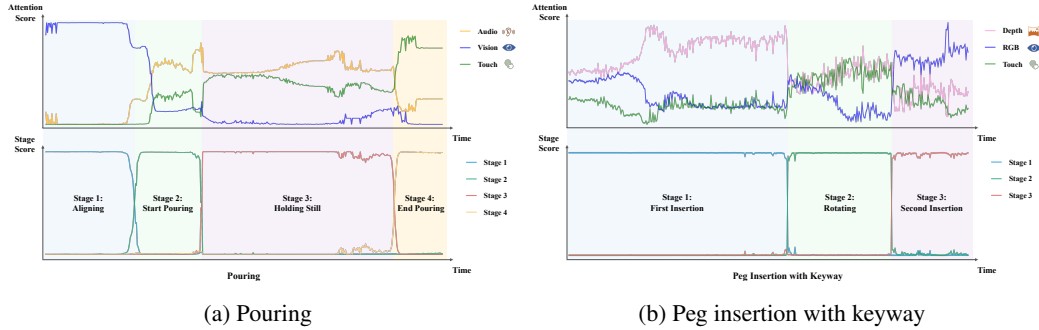

| (a) Pouring | (b) Peg insertion with keyway |

Figure 12: **Visualization of the aggregated attention scores for each modality and stage scores of MS-Bot in both tasks.** At each timestep, we average the attention scores on all feature tokens of each modality separately. The stage score is the softmax normalized output of the gate network.

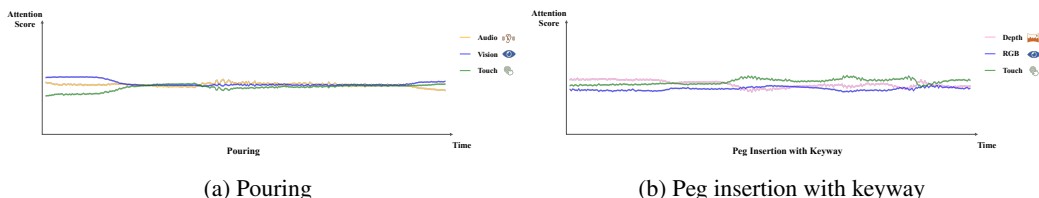

| (a) Pouring | (b) Peg insertion with keyway |

Figure 13: **Visualization of the aggregated attention scores of MULSA for each modality in both tasks.** At each timestep, we average the attention scores on all feature tokens of each modality separately. The range of the attention score axis in the figure is consistent with Fig. 12.

used to penalize the scores of the gate network in MS-Bot. We also attempt to modify the structure of the MULSA model to accept observation context as input. We input the observations from 5s, 7.5s, 10s, 12.5s, 15s, 17.5s, and 20s before the current timestep into the model, and encode these observation contexts by each modality's encoder into 7 tokens (per modality). These tokens are then concatenated with the original 6 tokens of the current timestep observation of each modality, resulting in a total of 13 tokens for each modality. We evaluate these baselines on the peg insertion with keyway task. The results in Tab. 5 shows that introducing an explicit stage prediction task could improve the performance of the Concat baseline, but it provides minimal benefit for the MULSA method. These baselines still have a performance gap compared to our MS-Bot. This indicates that explicit stage understanding is meaningful, but introducing context to aid stage understanding and specially designed stage-guided dynamic fusion modules are also necessary. In addition, the results show that inputting observation contexts into MULSA results in virtually no improvement in the model's performance. This indicates that mixing the contexts and current observations in self-attention calculations is highly inefficient for helping the model understand stages and allocate modality weights, as it reduces focus on the more important current observations and significantly increases GPU memory usage. In contrast, our MS-Bot leverages a long action history as context to enhance its understanding of task stages. This approach simplifies processing for the model and offers finer granularity, effectively guiding dynamic multi-sensor fusion.

# H   Comparison under the Multi-Camera Setting

In robotic manipulation, a common setup involves using multiple cameras to capture visual information from different views. By treating visual images from different views as distinct modalities, our MS-Bot is also able to handle multi-camera observations. To verify this capability, we conduct a comparison with baseline methods on the peg insert with keyway task under the setting with multiple cameras. Specifically, we place another camera on the side of the base, so that the lines connecting these two cameras to the base roughly form a right angle. We replace the depth observation from the first camera with the RGB observation from the second camera, as the RGB observations

| Method | Success Rate |
|---|---|
| Concat | 7/15 |
| MULSA | 11/15 |
| Concat + Stage Auxiliary Task | 9/15 |
| MULSA + Stage Auxiliary Task | 11/15 |
| MULSA + Observation Context | 11/15 |
| MS-Bot | **13/15** |

Table 5: Comparison of performance with stage-aware and context-aware baselines on the peg insertion with keyway task. We introduced stage prediction auxiliary tasks and multi-sensory observation context into the Concat and MULSA baselines for a more comprehensive comparison.

| Method | Success Rate |
|---|---|
| Concat | 9/15 |
| MULSA | 12/15 |
| MS-Bot | **14/15** |

Table 6: Comparison under the setting of multiple cameras on the peg insertion with keyway task.

from these two cameras already provide sufficient spatial information for this task. The results in Tab. 6 show that in the multi-camera setup, our MS-Bot method still outperforms the baselines. This demonstrates that our proposed stage-guided dynamic fusion can also be applied to fuse features from multiple cameras.

# I  Combination with Audio Modality

| Method | Success Rate |
|---|---|
| Concat | 8/15 |
| MULSA | 11/15 |
| MS-Bot | **13/15** |

Table 7: Comparison on the peg insertion with keyway task using RGB, depth, tactile and audio.

For tasks involving using lock-picking tools to unlock in real life, audio can indeed serve as the basis for completing the task, as these locks typically have specific mechanical structures that produce distinct sounds. Similarly, audio can indicate whether the key and keyway are aligned, thereby assisting with the peg insertion with keyway task. To verify the effectiveness of our model after further incorporating audio, we conduct experiments on the peg insertion with keyway task using RGB, depth, tactile and audio modality. The results in Tab. 7 show that the addition of the audio modality only brings improvement to the weakest Concat baseline. For MULSA and our MS-Bot method, the benefits are minimal. This is because the audio in this task is short and minimal due to the 3D-printed materials and is vulnerable to noise disturbance.

# J  Detailed Experimental Results of Pouring

In Sec. 4.4 and 4.5 of the main paper, we show the overall error on all settings of the pouring task. In this section, we comprehensively present the detailed result of the component ablation and the distractor experiment on all the settings of the pouring task in Tab. 8 and 9. As shown in Tab. 8, the contributions from both the state tokenizer and the stage comprehension module on all settings demonstrate the importance of the coarse-to-fine stage comprehension. The consistent lead of MS-

| Method | Pouring Initial (g) | | Pouring Target (g) | |
|---|---|---|---|---|
| | 90 | 120 | 40 | 60 |
| MS-Bot | **1.60 ± 1.10** | **5.58 ± 1.79** | **6.48 ± 1.55** | **1.80 ± 0.95** |
| - Attention Blur | 1.72 ± 1.09 | 5.70 ± 1.74 | 6.55 ± 1.38 | 1.95 ± 1.32 |
| - Stage Comprehension | 2.52 ± 1.12 | 6.15 ± 1.64 | 6.80 ± 1.59 | 2.92 ± 1.40 |
| - State Tokenizer | 3.05 ± 1.01 | 6.42 ± 1.98 | 7.12 ± 1.66 | 4.19 ± 1.24 |

Table 8: Impact of each component of our framework in the pouring task (mean ± standard deviation). '-' indicates further removing the module from the model in the previous line.

| Method | Pouring Initial (g) | | Pouring Target (g) | |
|---|---|---|---|---|
| | 90 | 120 | 40 | 60 |
| Concat | 8.75 ± 2.02 | 10.69 ± 2.45 | 10.72 ± 2.35 | 8.46 ± 2.03 |
| Du et al. [63] | 8.51 ± 1.79 | 9.54 ± 2.10 | 9.98 ± 2.20 | 8.04 ± 1.85 |
| MULSA [11] | 4.72 ± 1.30 | 7.83 ± 1.71 | 8.45 ± 1.50 | 5.56 ± 1.13 |
| MS-Bot | **2.04 ± 1.40** | **6.10 ± 1.49** | **7.62 ± 1.94** | **2.41 ± 1.47** |

Table 9: Comparison of performance in scenes with visual distractors in the pouring task(mean ± standard deviation). We change the color of the cylinder from white to red during testing.

BOT across all settings of the pouring task in Tab. 3 also demonstrates the generalizability of stage comprehension.

## K   Evaluation of Hyper-parameter Settings

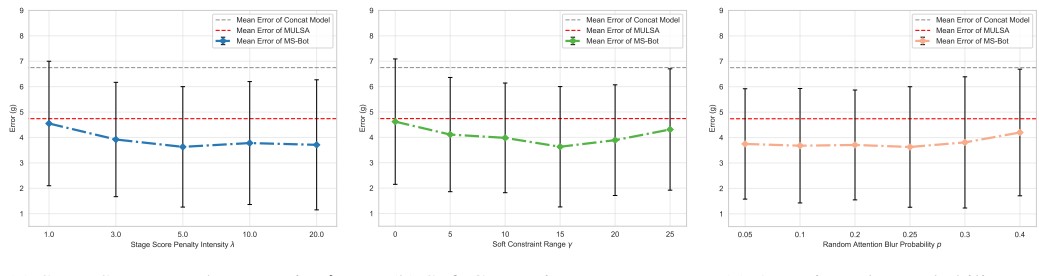

(a) Stage Score Penalty Intensity $\lambda$    (b) Soft Constraint Range $\gamma$    (c) Attention Blur Probability $p$

Figure 14: **Performance of MS-Bot on one setting of the pouring task when changing the hyper-parameters $\lambda$, $\gamma$ and $p$.** We fix the others when changing one hyper-parameter. We report the mean error of the concat model, MULSA and MS-Bot. The error bars represent the standard deviation of errors for MS-Bot.

In this section, we test the sensitivity of MS-Bot to different hyper-parameter settings. Specifically, we test the performance of MS-Bot under different penalty intensity $\lambda$, soft constraint range $\gamma$ and probability of random attention blur $p$ in one setting of the pouring task. We train all the models to pour out 40g of small beads with different initial masses (90g/120g). We set $\lambda = 5.0$, $\gamma = 15$ and $p = 0.25$ by default, and keep the others fixed when we modify one of the hyper-parameters.

We present the evaluation results in Fig. 14. Our MS-Bot consistently outperforms both the concat model and MULSA across various hyper-parameter settings, indicating that the performance of our MS-Bot is relatively stable to hyper-parameter variations. However, we also observe that when setting $\lambda$ and $\gamma$ to small values ($\lambda = 1.0$ in Fig. 14a and $\gamma = 0$ in Fig. 14b), the performance of MS-Bot drops and becomes closer to MULSA with simple self-attention fusion. This is because when $\lambda$ is too small, the prediction of the current stage becomes inaccurate due to the insufficient training. When $\gamma$ is too small, the model is forced to make drastic changes in the stage score prediction within very few timesteps, leading to unstable stage predictions. Both of these factors weaken the efficacy

of stage comprehension within MS-Bot. We also find that setting too large $p$ ($p > 0.3$ in Fig. 14c) can bring negative impacts as the attention blur truncates the gradients backpropagated to the stage comprehension module and the state tokenizer. Excessive attention blur can impair the training of these two modules.

## L   Limitations and Future Improvements

In this section, we discuss the potential limitations of our work and the corresponding solutions:

**In many robotic tasks this kind of dynamic weighting is not necessary.** Indeed, there are many tasks that can be completed with just the visual modality, where readings from other sensors may not provide useful information. We believe that one solution is to let the model first select the necessary subset of sensors based on the task (possibly using an LLM), and then perform dynamic multi-sensor fusion based on stage understanding. For tasks with only one stage, this can be considered as a special case where $|S| = 1$.

**The stage labeling may be costly.** Currently, our strategy for stage labeling is to segment the action sequence of the trajectory based on pre-set rules. This requires time to set up detailed rules and becomes more challenging when there are more stages and more complex stage transitions. A possible solution is the LLM-based automatic labeling method mentioned earlier, although a small amount of human effort is still necessary, such as constructing prompts to accurately describe the task setup and providing example for in-context learning.

**The lack of comparison to larger models.** Currently, there is no well-trained open-source multi-sensory large foundation model for robot manipulation (including vision, hearing, touch, etc.). Therefore, we are temporarily unable to verify the role of explicit stage understanding in large models, nor can we directly compare our model with multi-sensory large models. However, we believe that even for larger models, explicit stage understanding remains meaningful when introducing new modalities and training is required using a small amount of paired multi-sensory data, or when fine-tuning in new scenarios with limited data scale.

**Robustness issue in the case of missing modalities.** Since in both of our tasks there are stages that depend on a single modality, masking uni-modal features during testing may directly lead to failure. This is a problem common to all multi-sensory models, and we did not make improvements specifically for this issue. In cases where multiple sensors complement each other, the absence of uni-modal features may have a compounding effect on stage understanding and dynamic fusion. We believe that one possible solution is to randomly dropout uni-modal features during the training process [69]. Some other methods for robust multi-modal learning could also be referenced.

**Transferability of the current dynamic multi-sensory fusion architecture.** Since the multi-sensory fusion architecture we use in the dynamic fusion module is based on cross-attention rather than self-attention, there are difficulties in directly transferring the original form of this fusion method to transformers with multiple self-attention blocks. One potential strategy is to use the stage token as a special token to replace the class token. It could be used to predict the stage under the supervision of stage labels and thereby achieving the effect of stage-guided multi-modal fusion.

**Low integration level of current multi-sensory robot system.** Our current multi-sensory robot system has not well integrated the various sensors: the camera is positioned at a third-person perspective, the microphone is placed on the desktop, and the tactile sensor is at the end of the robotic arm. They are located in different positions. This low level of integration means that the system can only complete relatively simple tasks, while more complex multi-sensory tasks that require mobility cannot be accomplished. This multi-sensory system still needs further refinement and upgrading.

