# OpenReview forum: "Play to the Score: Stage-Guided Dynamic Multi-Sensory Fusion for Robotic Manipulation"
_robot-learning.org/CoRL/2024/Conference — CoRL 2024_

### Official Review · Reviewer_mfxN · 2024-07-19
**stage-integrated multisensory model; challenging manipulation tasks; ablation studies by masking different modalities**

**Originality:** 4
**Technical Quality:** 4
**Clarity Of Presentation:** 4
**Potential Impact:** 3
**Recommendation:** 4
**Confidence:** 5

**Review:**

The paper addresses a challenge in robotic manipulation with multisensory information. They introduce their ideas by presenting analysis of low confidence in action predictions during stage changing. Then, they propose a new method that take stage into consideration. The experiments are well-designed and analysis of the results are complete. The idea of their method is novel and original, and their paper is well-structured. The concept is quite inspiring and has is significant in multisensory robot learning.

Strength:
- An intuitive analysis on confidence in action prediction by masking out different modalities
- Propose a novel method for multisensory fusion with stage comprehension
- Challenging tasks design
- A comprehensive analysis of the experiment results to demonstrate performance improvement and significance of the stage module.

Weakness:
- The stages are pre-defined, so as for the stage feature dimensions. It might be not very ideal in real-world scenarios, but I can imagine that LLMs can be used to help label the stages.

**Quality Of The Limitations Section:**

2

**Questions For Rebuttal:**

- Table I pouring results is a bit confusing, which of initial weight corresponds to which target weight?
- For the key task, why do you decide not use audio in this setup? I can imagine that when opening a door lock, the sound actually help me get a sense if my key is correctly inserted. Also, people uses audio feedback when they use those lock picking tools to unlock doors.
- Could you elaborate more on how you will handle long-horizon tasks with more stages?

**Robotics Focus:**

4

**Summary Of Paper:**

The authors propose MS-Bot, a novel stage-guided multi-sensory fusion method with coarse-to-fine stage comprehension. Stage modules are designed to encode course-grained and fine-grained states. Authors demonstrate performance increase in pouring and insertion tasks.

**Summary Of Recommendation:**

The paper presents a novel method and complete analysis, and is clear to read. I recommend accept this paper.

---

### Official Review · Reviewer_ConB · 2024-07-21
**MDMP: Multimodal Diffusion for Motion Prediction with uncertainty in Human-Robot Collaboration**

**Originality:** 3
**Technical Quality:** 4
**Clarity Of Presentation:** 5
**Potential Impact:** 3
**Recommendation:** 4
**Confidence:** 4

**Review:**

The quality of this work is excellent, beyond the clarity of writing, the authors posit the interesting challange and an intuitive, well thought through solution. However I do have reservations about the performance benefit of the presented system in comparison to large multi-modal models, I would expect the blending, tuning and focus of individual modalities to be equally present in an end-2-end solution. Without this kind of comparison it is difficult to see if the method proposed beats state-of-the-art multi-modal models because it is actually essential in multi-modal robot systems to explicitly calculate stage state and weigh the multi-modal fusion based on this or if its because the current state-of-the-art methods (Du et al. and MULSA) are quite small models and require this explicit stage comprehension but larger models can handle it internally. Maybe there are reasons for the proposed approach from the perspective of demonstration dataset size conditions that can keep the impact of this work high. One method may be to compare this work against multi-modal versions of robot foundation models like Octo and OpenVLA, I understand this maybe a large effort/engineering challange to get those working, but it will validate the "stage comprehension" and "state tokenizer" aspects of this work which are the key novelties. The significance of the work is a little hard to picture because of the point raised above, however I think the research domain carved out in the introduction and related works is wide enough that this method and the challange presented in this paper of active modality weighting may become important in the future of robotics and I am excited to see where this research goes in the future, so for these reasons I think this work could have a strong impact. One barrier to high impact may be the requirements to label datasets with sub-task classes, to my knowledge the majority if not all of the common large robotics datasets do not contain this feature, and so larger implementations of this work may require significant efforts to enable large scale training.

The introduction is very well written, simple to understand and the authors present the ideas and intuition behind their work very well. The work is motiveated cleanly and the contributions are explained concretely. The related works is written equally well and covers enough ground to give the reader enough context to understand the work. This clarity continues throught the work and it was a joy to read (well done authors!)

List of Strengths:
- motivation for the work is clear and concise, with up to date referencing and context.
- They propose the interesting idea of "Modality Temporality", the challange of switching and focusing on different modalities for different robotics tasks.
- Contributions are clearly and concisely stated.
- The method is novel, intuitive and presents a nice pocket (active modality weighting) for research in the future.
- The method description is extremel well written, its clear, simple, the Figure 2 which describes the system coveres nearly all aspects of the method and pairs very well with the Section 3.2.
- The choices behind the proposed systems design is backed up throughout the methodology section and is easily understandable to the reader.
- The experimental section clearly states the key questions they try to answer about their proposed system. The questions are to the point and focused on the core questions I thibk readers will have about the proposed system.
- Experiments are made in the real-world with real hardware and applied to classic robotics benchmarks (peg in hole and pouring) and the visual distraction tests are a nice way to show benefit of the proposed method in comparison to SOTA multi-modal appraoches.
- The video was clear and concise. Speach overlay was really appreciated and made the video much more understandable. It may be worth editing so that the live weight during the pouring task is shown in a more readable way.

Weaknesses:
- As stated above, the issues come down to comparison with larger models that may make this kind of explicit sub-task comprehension redundant.
- limitations of the work are lacking, I think a more comprihensive analysis of the work would really help readers understand where this work is currently failing and can be improved. Some limitations to address maybe that in many robotics tasks this kind of weighting is not necessary, the cost of labelling demonstrations with sub-task class, comparison to larger models, how this trained system may lack the robustness of an integrated system when dealing with uni-modality occlusions.

Overall I think the paper is of high quality and well suited to CoRL. The authors present a novel task (Modality Temporality) and a strong performing solution to this task, experiments show both the reason this presented task are important and the strengths of their proposed solution. Given the intuition and results presented, this task may become an area of interesting research in the future. The work lacks comparison to a model large enough to potentially perform sensor modality weighting in an end-2-end manner that may make the task and therefor solution redundant. However, that being said I think that comparison would likely need to be a seperate paper due to it's scale. A great paper and a joy to read.

**Quality Of The Limitations Section:**

1

**Questions For Rebuttal:**

Section 1:
- Line 23: need to cite or show that optimus and figure 1 are multi-sensory robots.
- Line 43-46: Is the general sentiment of this sequence correct? Do the later layers in sensory fusion methods not weigh the given sensor features naturally? I would expect this to be the case within the context of an end-to-end system. I am open to discussion on this but my intuition was that end-to-end systems would learn to weigh individual modalities during the course of a multi-stage task, expecially if enough observational context is given to understand the task stage (it would have to understand the task stage to complete the multi-stage tasks), some logic behind your call for explicit stage understanding within the context of sensor modality weighting, in comparison to an end-2-end approach with enough context to know what stage it is in, would help motivate this research to the reader.

Section 2:
- Clear concise and well written, giving the reader enough context to understand the proposed solutions and research space. I have no comments that need addressing for this section.

Section 3:
3.1:
- Line 106-107: What is the reasoning behind not using previous sensor readings in this method, is it to keep the model simple or is there a more complex reason behind this decision? A little sentence here would be useful for readers to understand the design decisions behind the model.
- Line 112-121: Again this is very well written and makes perfect sense to me, I once again wonder if your intuition is correct. You state "could" so there is nothing wrong with this paragraph at all, however it "could" be improved if you had some backup evidence for your intuition, the results show that this method out performs MULSA which is end-to-end so you can use this to clarify to the reader that this intuition is currently correct, but given the lack of context in MULSA it is difficult to explore if this weighting would occure with multi-modal robot foundation models which take a context sequence of observations as input.

3.2:
- You could add z^{fus}_{t} to Figure 2 so that the figure better aligns with the Dynamic fusion description (Line: 171,172)

Section 4:
- Implementation of the methods details would be good to know, barebones essential parameters for the architecture and the training. Especially if there are querks to getting this system trained well.
- Would be good to get more details on the 1st baseline ( the concat model which directly concatenates all the uni-modal features) - is it literally just this or are there other aspects to this?
- I would like to see a comparison with a larger end-to-end solution that may make explicit tuning of modality weighting redundant, specific focus on the dynamic modality focus would be a great way to motivate the necessity to solve and explore the presented "Modality Temporality" challange and invite research in this area, helping produce a larger impact for this work. Although I understand this maybe a large undertaking and I do not think it is essential for this paper to be accepted. I think the best version of this paper would show that large models will still require this explicit sub-task modality weighting.
- I would like to see a comparison of attention scores between the MULSA method and the method proposed in this work. That would provide insight into the level of impact the explict stage-based dynamic fusion is having.

Section 5:
- Limitations of the work are not discussed at a high enough standard, I think a more comprehensive analysis of the works limitations would really help readers understand where this work is currently failing and can be improved. Some limitations to address maybe that in many robotics tasks this kind of weighting is not necessary, the cost of labelling demonstrations with sub-task class, comparison to larger models, how this trained system may lack the robustness of an integrated system when dealing with uni-modality occlusions. I'm open to discussion on whether any of these are in fact limitations. Likewise I'm sure there are more to add.

**Robotics Focus:**

4

**Summary Of Paper:**

The paper explores the how to adaptively weigh individual modalities ("modality importance") of a multi-sensory robot when handling mutli-stage tasks within the context of complex sub-goals that require different levels of focus on given modalities. The core intuition is that sensory fusion needs to change for the given sub-task to be the most effective. They build a multi-sensory robot and use imitation learning to train a MULSA model, they additionally add a method to identify task stage and use this classification to adjust modality weights. The proposed method (MS-Bot) is applied to a pouring and a peg insertion with keyway task.

**Summary Of Recommendation:**

Overall I think the paper is of high enough quality for CoRL. The authors present a novel task (Modality Temporality) and a strong performing solution to this task, experiments show both the reason this presented task are important and the strengths of their proposed solution. Given the intuition and results presented, this task may become an area of interesting research in the future. A great paper and a joy to read.

---

### Official Review · Reviewer_rczR · 2024-07-26

**Originality:** 2
**Technical Quality:** 3
**Clarity Of Presentation:** 3
**Potential Impact:** 2
**Recommendation:** 3
**Confidence:** 4

**Review:**

Strengths
- Interesting idea to incorporate stage information for mixing the information from multi-modal sensors
- Ablation studies that show the importance of each component
- Writing is mostly clear and helps understanding the main idea
- Real-robot experiments

Weaknesses
- The proposed idea heavily depends on the additional information about the stage within the episode. This could be very difficult to obtain in practice. Would there be any alternative way of extracting such stage information from the data without human annotation? Any preliminary experiments can strengthen the claims made in the paper.
- Baselines don't have an access to additional stage information, which makes it difficult to think that the results are from fair comparison. For instance, very naive baseline that incorporates stage information by having an auxiliary objective that predicts the stage can be a useful baseline to have.
- One of missing baselines is that training DiffusionPolicy with multimodal features, as its transformer network can in principle learn to focus on useful features via self-attention. And investigating what happens when we use more expressive policy network with multi-modal inputs can be helpful for strengthening the claims made in the paper.
- One missing experimental setup that could be important is how it handles features from multiple cameras -- which is more of a common setup than using audio or tactile sensors.
- It could be nice to add more results that show how the proposed method is sensitive to the hyperparameters.
- More formal definition or decsription of states and stages could be provided in the initial part of Method section,  with clear description on how a stage label is added in line 131.
- What does 'stage after warmup' mean in line 147?
- Motivative experiments in Introduction needs to be more self-contained as there is no explanation about what confidence means. and how it is computed.

**Quality Of The Limitations Section:**

2

**Questions For Rebuttal:**

- See Review

**Robotics Focus:**

4

**Summary Of Paper:**

This paper presents an imitation learning method that learns to fuse features from different modalities using additional information about the stage within the task. In particular, action history and current features are used to extract current state and stage, and this is information is used to weight the multimodal features, and weighted features are used for predicting actions. Experiments show that the proposed method outperforms baselines that don't explicitly consider such stage information.

**Summary Of Recommendation:**

I enjoyed reading the paper but the proposed idea is a bit limited in that it heavily depends on expensive stage annotation, and the experiments are missing some baselines and analysis that could further strengthen the claims made in the paper.  I'm currently in a borderline but slightly leaning towards weak reject. But I'm willing to change my score after reading other reviewers' opinions and the rebuttal response.

---

### Author Rebuttal · Authors · 2024-08-11

We would like to express our sincere appreciation to all reviewers for their insightful and comprehensive feedback. We sincerely appreciate the comments that the idea is interesting (Reviewer rczR, ConB and mfxN), the presentation is clear (Reviewer rczR, ConB and mfxN), the real-world experiments are well-designed (Reviewer ConB and mfxN) and the concept of "Modality Temporality" is inspiring (Reviewer ConB and mfxN).

In this short note, we summarize the main concerns about our work and our corresponding response:

[Stage labeling without human annotation] Extracting stage information without pre-defined stage division and human annotation is feasible. We present the approach to using LLMs for automatic stage annotation in the **attached rebuttal file**. By constructing prompts to accurately describe the task setup and providing example action sequence, we can utilize LLM to automatically divide the task into different stages and assign stage labels to each action in the action sequence.

[Significance in large models] Introducing explicit stage understanding as prior knowledge could still be meaningful for the multi-sensory large models, especially when we wish to introduce new sensors and fine-tune large models with relatively small amounts of new sensor data.  The implicit task stage understanding that needs to utilize information from the new sensor could remain suboptimal due to the limited scale of data of the new sensor. Therefore, we believe that explicit stage understanding and stage-guided dynamic multi-sensory fusion still hold research significance in large models.

[Additional baselines] We have supplemented experiments comparing with more baselines including:

- Baselines including stage auxiliary task (Reviewer rczR) (Meanings of specially designed stage-guided dynamic fusion module)
- DiffusionPolicy (Reviewer rczR) (Applicability to more expressive policy network)
- Baselines in multi-camera settings (Reviewer rczR) (Applicability to the more common multi-camera setup)
- MULSA with observation context (Reviewer ConB) (Meanings of explicit stage understanding and specially designed stage-guided dynamic fusion module)
- Baselines additionally using audio modality (Reviewer mfxN) (The reason for the choice of modalities for the peg insertion with keyway task)

**We have attached a rebuttal file introducing the automatic stage labeling method using LLM.** We present examples of automatic stage division and stage labeling by LLM in the file.

---

### Decision · Program_Chairs · 2024-09-04

**Decision:**

Accept

**Comment:**

Strengths:
- Clear motivation and presentation;
- Innovative idea;
- Real-world results.

Weaknesses:
- The idea may not be timeless as larger models evolve;
- The stages need to be pre-defined;
- Some baseline comparisons may be included.

Post rebuttal: The authors did an excellent job during the rebuttal, after which all three reviewers recommended acceptance, including two strong accepts. The AC agrees with the recommendation. The authors are encouraged to revise the submission to clearly acknowledge the limitations, such as pre-defined stages, in the camera-ready.